# The Latent Class Analysis of Adverse Childhood Experiences among Chinese Children and Early Adolescents in Rural Areas and Their Association with Depression and Suicidal Ideation

**DOI:** 10.3390/ijerph192316031

**Published:** 2022-11-30

**Authors:** Chun Chen, Yu Sun, Boyuan Liu, Xiao Zhang, Yingquan Song

**Affiliations:** 1School of Humanities and Social Sciences, Chinese University of Hong Kong-Shenzhen, Shenzhen 518172, China; 2Department of Education Policy Studies, Pennsylvania State University, State College, PA 16801, USA; 3Department of Sociology, Tsinghua University, Beijing 100184, China; 4China Development Research Foundation, Beijing 100011, China; 5China Institute for Educational Finance Research, Peking University, Beijing 100871, China

**Keywords:** adverse childhood experiences, latent class analysis, depression, suicidal ideation, Chinese rural areas

## Abstract

Exposure to adverse childhood experiences (ACEs) is a global public health concern that is detrimental to the psychological outcomes of Chinese children in rural areas due to the lack of public awareness of ACEs and mental health resources. The objective of this study was to identify the patterns of ACEs and the impact of ACE patterns on depression and suicidal ideation among 4683 students (mean age = 10.08 years, SD = 0.99; 48.17% female students) from 63 elementary schools in rural areas in Guizhou Province, China. Latent class analysis was conducted to identify the best class pattern. A three-step approach was undertaken to explore the association between the class patterns and demographic covariates and depression and suicidal thoughts. An overall three-class pattern of ACEs was identified, which was: (1) high ACEs, (2) high verbal abuse and emotional neglect and low household dysfunction, and (3) low ACEs. The results also showed that children in the high ACEs class tended to show higher depression rates and more frequent suicidal ideation across the three groups. Being female and younger and having a lower socioeconomic status were risk factors. Our study identified a class pattern that was not found in previous research, which is high verbal abuse and emotional neglect and low household dysfunction.

## 1. Introduction

Adverse childhood experiences (ACEs) have been documented as an alarming public health concern [1]. ACEs are defined as exposures to chronic stressors or potential trauma events that are experienced by the individual as physically or emotionally harmful, threatening, or overwhelming before the age of 18 and that have lasting and holistic effects on individuals’ functioning [2]. Some of the most common ACEs include physical abuse, verbal abuse, sexual abuse, physical neglect, and household dysfunction [2]. With the increasing awareness and advocacy of child’s rights in Chinese society, there has been a recent burgeoning of research on family ACEs among Chinese children, revealing a shockingly high prevalence rate of ACEs in China. It was reported that around 66% to 79% of the Chinese child population, including samples from elementary school students in Bengbu, Anhui Province [3], junior high school students in Shanghai [4], and rural high school students in China [3], had experienced at least one ACE. Around 21.5% of elementary school students in Anhui Province reported experiencing four or more ACEs [5]. In comparison, 10% of junior high school students in Shanghai reported experiencing five or more ACE [4]. Among different ACEs, physical abuse and domestic violence [3] were reported to occur the most among the Chinese population. However, most of the above studies were conducted in urban areas in China. The acknowledgment of ACEs in rural Chinese regions is important due to their detrimental impacts on physical and mental health outcomes [6]. Due to the high proportion of left-behind children (i.e., children where one or both parents moved to urban areas for work and the child stayed in rural areas; [6]) as well as the lack of awareness to protect children in rural China [7], the risk of ACE occurrence was found to be higher among rural children than their urban counterparts [6]. Meanwhile, there is a lack of a person-centered approach to examining the profile of ACEs among rural students. Therefore, the present study aimed to provide a person-centered exploration of ACEs among children and early adolescents in rural Chinese areas and their relationship with depressive symptoms and suicidal ideation. Our research aimed to help give implications for when practitioners work with Chinese rural area students with the goal of early mental health prevention and intervention.

### 1.1. Latent Class Analysis of Adverse Childhood Experiences (ACEs) in China

Previous studies have mainly focused on using a variable-centered approach to understanding individual or cumulative ACEs [7]. However, a variable-centered approach cannot reflect the nuanced experiences of individuals with the same ACE score [8,9]. Therefore, in the past two years, there has been a burgeoning trend of using a person-centered approach to examine ACEs in China, such as latent class analysis (LCA). We identified approximately ten studies that used LCA in a Chinese context (e.g., [5,10,11]). For example, in one study conducted among 433 Chinese young adults in Hong Kong, three patterns were identified: low ACEs, household violence, and multiple ACEs [10]. Another study among 1766 elementary school students in Anhui Province found four patterns: high ACEs, highly abusive and adverse events, highly abusive and neglected, and low ACEs [5]. Overall, most studies found similar patterns of low ACEs, high ACEs/multiple ACEs, and classes in the middle reporting varied severities of different types of ACEs among young Chinese populations. The prevalence rates of class patterns were also similar. In general, the low ACEs class accounted for the highest percentage, 50% to 70% (e.g., [5,10]. People classified with high ACEs or multiple ACEs accounted for the lowest percentage of 5% to 20% (e.g., [5,10]). However, we were only able to identify one study that examined patterns of childhood trauma among rural-to-urban migrant children in Beijing, one of the biggest cities in China [12]. There is still a considerable lack of research on using a person-centered approach among children and adolescents from under-resourced rural areas in China. Understanding ACEs among rural students is of importance. Children in rural areas in China tend to be from relatively socially and economically disadvantaged families. For example, left-behind children are commonly observed in rural areas [13]. The growing population of left-behind children is a huge family issue in rural China [14]. Due to the high proportions of the left-behind phenomenon, children and adolescents in rural areas receive less parental involvement in their development [15]. Presumably owing to these reasons, rural students suffered from more ACEs. More importantly, profiles of ACEs in rural students may differentiate from profiles of their urban counterparts.

### 1.2. ACEs and Psychopathology

Depression is a common psychological disorder that impairs an individual’s emotional and cognitive development. Among environmental risk factors of depression, traumatic experiences, particularly exposure to ACEs in early life, have been found to play a significant role in the development, course, and maintenance of depression [16]. For example, according to a meta-analysis of 65 articles on the relationship between ACEs and mental health, on average, those with ACEs were twice as likely to suffer from depression [17]. People with a higher number of ACEs had more depressive symptoms [18]. The accumulation of ACEs was a significant predictor of depression, and how early an individual first experienced the childhood trauma was also found to relate to the possibility of depression. First exposure to ACEs in early childhood (ages 0–5) was more strongly associated with depression in early adulthood than ACEs later in life [19]. Furthermore, different types of ACEs were also found to predict depression in different magnitudes among various samples. Compared with neglect and family dysfunction, childhood abuse had the most substantial contribution to depression, according to an extensive systematic literature review study [17]. However, in another meta-analysis conducted on childhood trauma and adult depression studies, emotional abuse in childhood showed the strongest association with depression in adulthood, followed by neglect, sexual abuse, domestic violence, physical abuse, parental divorce or separation, and child hospitalization [20].

According to the dose–response theory [21], the magnitude of ACEs was also found to be a significant predictor of depression severity. For example, children with high ACEs were at higher risk for depression in China [10]. Lee et al. [22] found that the multiple high-risk class had the most prominent depression rate, followed by the multiple low-risk class, the broken family class, and the income hardship class among children and adolescents in the U.S. Blum et al. [23] conducted a comparative study that proved the strong association between ACEs and depressive symptoms in all 14 countries, and the class with high exposure to neglect and physical and emotional abuse was more likely to have depressive symptoms than high exposure to violence victimization and household instability. Another LCA study conducted among university students in East Asia found that individuals in the household violence class had significantly higher depression than those in the low ACEs class [19].

Although previous epidemiological evidence shows a strong dose–response link between ACEs and depression, the depression rates are significantly higher among students in rural areas in China than those in urban cities (e.g., [24,25]: roughly 23% of rural children exhibit symptoms of depression compared with only 13.5% of urban children) and the negative impact of ACEs on adolescent depression in China [7,18], it became critical to examine the association between ACEs and depression in this understudied population in China.

Child suicidality is a growing public health concern worldwide [26]. A growing body of research has pointed out the significant impacts of childhood factors on children’s suicidality rate (e.g., [27]. In particular, public health research has demonstrated ACEs as one of the significant predisposing factors for probability of suicidal behaviors [27], which was confirmed among 1265 children in New Zealand [28] and 5692 samples in the U.S. [29]. The negative impact of ACEs on child and adolescent depression and suicidality was also observed in China [7,18,30].

Among all types of ACEs, childhood abuse, including sexual, physical, and emotional, are the most critical contributors to suicidal risk (e.g., [29]). For example, parents’ divorce was found to predict children’s suicidal behaviors in rural China [24]. Wan et al. found that ACEs were found to be associated with suicidal attempts among 14,820 adolescents from both urban and rural Anhui, Henan, and Guizhou provinces in China [30]. Based on 14,500 middle school students in large cities in China, there were four distinct ACE patterns: high ACEs, high abuse and neglect, high neglect, and low ACEs, and students with high ACEs have higher suicide risks [31].

### 1.3. The Present Study

The present study intended to fill the gap in the ACE literature to understand the ACE patterns among Chinese children and early adolescents in rural areas and their subsequent experiences of depression and suicidal ideation. First, we aimed to investigate the prevalence of ACEs. Second, most of the previous studies were conducted from a data-centered approach, and there was still a lack of research on rural Chinese children. Up to now, little research has focused on rural China areas and primary school students’ suicidal behaviors using the LCA method. In different types of dysfunctional families, adversities may happen in various combinations. Therefore, the LCA approach was conducted in the present study to understand the profiles of ACEs. Third, we investigated the associations between demographic variables (i.e., age, sex, ethnicity, socioeconomic status [SES], and paternal and maternal left-behind status) and ACEs class profiles. Fourth, we explored the associations between ACE class and depression and suicidal ideation outcomes.

## 2. Materials and Methods

### 2.1. Participants

Participants in this study were 6- to 15-year-old (mean age = 10.08 years, SD = 0.99) students in 63 rural primary schools in Guizhou Province, China. A total of 4683 students were recruited for the study, with an average number of 67 students and a standard deviation of 45.61 in each school (min = 7 students per school; max = 169 students per school). Upon data cleaning, 4548 students were included in the final sample. Therefore, the effective response rate was 97.12%. Participants’ sex and grade levels were well balanced, with 48.17% being female students and 32.13% being in 3rd grade, 32.99% in 4th grade, and 34.88% in 5th grade. The majority of the participants were of Han ethnicity at 86.62%. About half of the participants’ fathers and/or mothers had left them and moved to another city for better job opportunities, with 59.53% fathers who left and 45.55% mothers who left. The overall sample was from relatively lower familial socioeconomic status (SES) compared with typical middle-class families in China.

### 2.2. Data Collection

The present study was a part of a project that aimed to examine rural students’ psychological well-being and school functioning. Data were collected in October 2020. First, the research team established collaboration with the rural elementary schools in Guizhou Province, and 63 elementary schools agreed to participate in the project. Second, simple random sampling was conducted to select two random class cohorts from all third- to fifth-grade classrooms. Parents were required to provide passive consent to agree for their child to participate in the study, and students needed to provide their assent before participation. The survey was distributed via paper-and-pencil format. Because the survey contained items that might provoke psychological discomfort, we provided the free local mental health hotlines to participants and parents on the informed consent. The study was conducted in accordance with the Declaration of Helsinki, and the protocol was approved by the ethics committee (IRB00001052-20020) of one of the corresponding authors’ universities.

### 2.3. Measures

#### 2.3.1. Adverse Childhood Experiences (ACEs) Questionnaire

Adverse childhood experiences during the respondent’s life were measured by the Adverse Childhood Experiences (ACEs) questionnaire, which was originally developed by the U.S. Centers for Disease Control and Prevention [32]. The ACEs questionnaire is a 10-item checklist that measures ACEs across 3 dimensions: abuse (3 items: physical, verbal, and sexual), neglect (2 items: physical and emotional), and household dysfunctions (5 items: parental separation, domestic violence, drug use, family mental disorders, and family incarceration). Example questions included “Did a parent or other adult in the household often: swear at you, insult you, put you down, or humiliate you?”, “Did you often feel that: No one in your family loved you or thought you were important or special?” and “Did you live with anyone who was a problem drinker or alcoholic or who used street drugs?” Each item was coded dichotomously based on the presence (coded as 1) or absence (coded as 0) of experience in the student’s childhood. The sum of all affirmative answers represents the ACEs score. A higher score indicates a higher frequency of experiencing adverse events. The Chinese version of the ACEs questionnaire was translated and validated among Hong Kong clinical samples by Fung and colleagues [33]. Although previous studies have used part of the ACEs questionnaire among Chinese participants younger than 18 years old (e.g., [6,34]), the ACEs questionnaire was originally developed to conduct among participants older than 18 years old [33]. Thus, confirmatory factor analysis was conducted to verify if the factor model fits the present sample. Results confirmed that the scale demonstrated a one-factor model on the 10 items, which showed the same (*χ*^2^(33) = 185.293, *p* < 0.001, RMSEA = 0.032, 90% CI = 0.028–0.037, CFI = 0.94). Cronbach’s alpha was 0.71 for the ten items in the present study.

#### 2.3.2. Depressive Symptoms and Suicidal Ideation

The Center for Epidemiological Studies Depression Scale for Children (CES-DC) [35] comprises 20 standardized items to evaluate students’ depressive symptoms in the past week. The items include short and simple statements that tap into the emotional, cognitive, and behavior-related symptoms of depression. All items are evaluated on a 4-point Likert scale concerning their incidence during the previous week and scored from 1 to 4 (1 = not at all, 2 = a little, 3 = some, 4 = a lot). Higher sum scores indicated a greater number of symptoms. Sample items include “I was bothered by things that usually don’t bother me” and “I felt down and unhappy.” The psychometric properties of the Chinese version of the CES-DC have been empirically tested among Chinese elementary school students [31], showing good concurrent validity, excellent construct validity, and adequate internal consistency reliability. In the present study, the scale demonstrated a one-factor model in the sample on 16 items (*χ*^2^(104) = 1131.557, *p* < 0.001, RMSEA = 0.047, 90% CI = 0.044–0.049, CFI = 0.92). Four items were dropped due to low factor loadings. Cronbach’s alpha was 0.86. Following the CES-DC [35] questionnaire, an additional item was asked to measure students’ frequencies of suicidal ideation in the past week: “I have thought about attempting suicide. 1 = never to 4= often (5–7 times)”.

#### 2.3.3. Demographic Variables

Participants were asked questions about demographic measures, including their age, sex, ethnicity, SES, and left-behind status (paternal and maternal separately). Ethnicity was collected because China is a multiethnic country. SES was gathered by asking whether they have specific household appliances that fulfill basic needs (i.e., water, flushable toilet) or are commonly observed in a middle-class family in China (i.e., solar system, washing machine, TV, computer, Internet, refrigerator, fan), as well as transportation tools in rural Chinese rural areas (i.e., motorcycle, vehicle/van, tractor). Participants endorsed 0 = none or 1 = yes towards each item. The sum score constitutes the SES score. Paternal and maternal left-behind status was assessed by asking whether their father or mother migrated to another city for work and did not return home for more than half a year in the past year. Participants rated 1 = yes and 2 = no on the item.

### 2.4. Statistical Analyses

In order to identify patterns of ACEs among the participants and due to the nature of ACE items to be dichotomous, latent class analysis (LCA) was conducted to study the heterogeneity and identify the most mutually exclusive classes of 9 ACEs using Mplus 7 [36]. We began the LCA by fitting a one-class unconditional model and increasing the number of classes. Model fit was assessed based on multiple model fit statistics, including Akaike’s information criteria (AIC), Bayesian information criteria (BIC), and sample-sized adjusted BIC (saBIC); approximate weight of evidence criterion (AWE); Lo–Mendell–Rubin test (LMRT), bootstrap likelihood ratio test (BLRT), approximate correct model probability (cmP), and Bayes factor (BF). Masyn [37] suggested that lower values of BIC, CAIC, saBIC, and AWE indicates a better model fit. At the same time, the BF compares the fit between model K (i.e., the present model) and model K + 1 (i.e., the model with one more class). When BF was less than 3, model K + 1 was more valued than model K; when BF was greater than 10, model K was considered a better model [37]. In addition to the fit statistics, parsimony and substantive meaning of classes were also considered in the model fit [37].

Once the best model was established, the next step was to determine whether there were significant differences in auxiliary variables (i.e., covariates (age, gender, left-behind status, and SES) and distal outcomes (depression and suicidal ideation)) across the identified classes. The three-step method [38] was conducted to include covariates and distal outcomes [39,40]. Mplus was used for the analyses. The three-step approach is preferred because it helps to ensure that the latent classes are not influenced by covariates and distal outcomes [39,41]. The covariates (i.e., age, gender, SES, parental left-behind status) and distal outcomes (i.e., depression and suicidal ideation) were included using multinominal logistic regression. A reference class was chosen to compare the probability of a person being classified in a given class relative to the reference class. Meanwhile, mean estimates of the distal outcomes were compared between each latent class. Overall, maximum likelihood (ML) estimation was used to handle missing data in analyses.

## 3. Results

### 3.1. Descriptive Statistics: Prevalence of ACEs, Depression, and Suicidal Ideation

Overall, the rates of each ACE were as follows: childhood verbal abuse was 15.9%, physical abuse was 12.7%, sexual abuse was 13.8%, emotional neglect was 21.8%, physical neglect was 11.6%, divorce was 16.4%, domestic violence was 8.2%, family drug use was 6.2%, family mental disorder was 6.5%, and family incarceration was 8.9%. The percentage of students who reported experiencing at least one ACE was 51.1%, which indicated that roughly half of the students in the present study experienced at least one childhood trauma in their life so far. The mean of cumulative ACEs was 1.27 (SD = 1.74). From the correlation matrix shown in Table 1, apart from demographic variables, depressive symptoms and suicidal ideation were both positively correlated with cumulative ACEs, while depressive symptoms and suicidal ideation were also positively associated with each other.

### 3.2. Latent Class Profiles of ACEs

Each ACE item was treated as a categorical variable as to whether the participant had experienced harmful childhood exposure or not. The ten items are represented as verbal abuse, physical abuse, sexual abuse, emotional neglect, physical neglect, parental separation/divorce, witnessing domestic violence, family drug use, family mental disorder, and family incarceration. The fit indices from one-class to five-class models, as presented in Table 2, suggested selecting either a three-class, four-class, or five-class model. However, the three-class model was selected considering the optimal indices, parsimony, class meaningfulness, and model comparisons. The three classes were represented as (1) low ACEs, comprising 61.3% of the sample; (2) high verbal abuse and emotional neglect and low household dysfunction, comprising 22.5%; and (3) high ACEs, comprising 16.1%. Figure 1 presents the distribution of the three classes. In detail, the high ACEs class demonstrated a consistently high score across all the ACE items. High verbal abuse and emotional neglect and low household dysfunction class showed higher probability scores of abuse items, particularly those of verbal abuse and emotional neglect, and much lower probability scores of household dysfunctions, except parental separation, which was as high as sexual abuse and physical abuse. Last but not least, the all-low ACEs class showed the lowest probabilities for all items.

### 3.3. Demographic Factors Associated with Latent Class Profiles

The likeliness of being placed in a particular class differed among different demographic variables. As shown in Table 3, when using the low ACEs class as the reference group, participants who were female, younger, and in lower SES were more likely to be classified in the high ACEs class (sex: logit = 0.511, *p* < 0.001, OR = 1.61; age: logit = −0.326, *p* < 0.001, OR = 0.772; SES: logit = −0.145, *p* < 0.001, OR = 0.978). Compared with the low ACEs class, students who were female and younger were more likely to be placed in the high verbal abuse and emotional neglect and low household dysfunction class (sex: logit = 0.476, *p* < 0.001, OR = 1.668; age: logit = −0.259, *p* < 0.001, OR = 0.722).

### 3.4. Latent Class Profiles Associated with Depression and Suicidal Ideation

Overall, participants in high ACEs demonstrated the highest depression scores across the three classes. When comparing the means across the three classes, respectively, as shown in Table 4, the means of depression scores varied significantly. Participants in high ACEs (mean = 2.40) and participants in high verbal abuse and emotional neglect and low household dysfunction (mean = 2.18) were more likely to show higher depression scores than those in low ACEs (mean = 1.63; differences between class 1 and class 3: *p* < 0.001, *d* = 0.78; differences between class 2 and class 3: *p* < 0.001, *d* = 0.22). At the same time, participants in high ACEs (mean = 2.40) were more likely to have higher depression scores than those in high verbal abuse and emotional neglect and low household dysfunction (mean = 2.18; differences between class 1 and class 2: *p* < 0.001, *d* = 0.56).

A similar pattern of the mean differences in suicidal ideation frequency was demonstrated in Table 4. Suicidal ideation occurred most frequently among participants in the high ACEs class (mean = 3.43). When comparing the means across the three classes, the means of suicidal ideation frequency also varied significantly. Participants in high ACEs and participants in high verbal abuse and emotional neglect and low household dysfunction (mean = 1.29) reported more frequent suicidal ideation than those in low ACEs (mean = 1.06; differences between class 1 and class 3: *p* < 0.001, *d* = 2.37; differences between class 2 and class 3: *p* < 0.001, *d* = 2.14). At the same time, participants in high ACEs were more likely to have suicidal ideation than those in high verbal abuse and emotional neglect and low household dysfunction (differences between class 1 and class 2: *p* < 0.001, *d* = 0.23).

## 4. Discussion

To the best of our knowledge, this was the first study that assessed the profile of ACEs and the relationship between patterns of ACEs and depression and suicidal ideation among elementary school students in underserved rural areas in China. Among all the ACEs, the prevalence rate of emotional neglect was the highest (21.8%). A three-class pattern was identified among the participants, with (1) high ACEs, (2) high verbal abuse and emotional neglect and low household dysfunction, and (3) low ACEs. Differences in depression and suicidal ideation were found to vary across the three classes.

### 4.1. Latent Class Profiles of ACEs

Partially consistent with the previous LCA studies conducted among the Chinese population that identified ACE patterns (e.g., [5]), in a three-class model with high ACEs, high verbal abuse and emotional neglect and low household dysfunction, and low ACEs were classified based on the differing amount and severity of ACEs. Consistent with previous studies among Chinese young adults [10] and elementary school students in Anhui Province in China [5], our study also classified an overall low-risk profile and an overall high-risk profile. Unlike Ho et al. (2019), we did not identify a household violence class. Particularly, our results assimilated the most with Zhang et al.’s study [5]. Both identified a highly abusive and neglected class.

In the present study, participants from the low ACEs pattern account for 61.3% of the sample, consistent with previous studies of 50% to 70% participants in the low ACEs group [5,10]. Participants with high ACEs accounted for 16.1%, which was higher than the percentage of less than 10% in previous studies [5]. One unique finding of our research is that we identified a class pattern that was not found in previous research: high verbal abuse and emotional neglect and low household dysfunction. The high verbal abuse and emotional neglect and low household dysfunction class comprised 22.5% of the entire sample, about one in five students in rural Chinese areas. Participants in this group reported more physical abuse, emotional neglect, and verbal abuse and less sexual abuse and household dysfunction. The existence of this group is likely due to the fact that compared with the awareness of classifying sexual abuse as ACEs, there was a lack of awareness of considering verbal abuse, physical punishment, and emotional neglect in Chinese society. Therefore, even when there was no family dysfunction or risky family environment, these types of abuse are still commonly observed in Chinese households.

### 4.2. Demographic Factors Associated with Latent Class Profiles

Our results found that compared with students in low ACEs, students who were female and younger were at a higher risk of being classified in high ACES and high verbal abuse and emotional neglect and low household dysfunction class. In contrast, students who were in lower SES were more likely to be in high ACEs. Regarding the sex differences, contradictory to some of the previous ACEs studies in China [42], female sex was found to be a risk factor for ACEs among rural Chinese children. It is likely that in rural areas in China, the patriarchal phenomenon prevails in Chinese families, and female children were historically underserved, which made them receive less attention and more likely to receive maltreated parenting from their families [43]. Female children are likely to be more sensitive and have stronger awareness of their stressful experiences at home [44]. At the same time, different from some of the previous ACE studies in the U.S. (e.g., [45]), the current finding showed that younger age was a risk factor for ACEs. Participants who were at a younger age were likely to have less power over how their families treated them and therefore were more vulnerable to exposure to ACEs. It is also likely that younger elementary school students might not have fully comprehended the items, which resulted in elevated responses. However, it is also possible that relatively older respondents might not feel safe enough to disclose their ACEs due to social desirability or not wanting to lose face for their families. Furthermore, coming from a lower SES background was also a pronounced risk factor for students to experience high ACEs. It is likely that parents of respondents who came from a low SES background are less educated on how to create a safe and healthy family environment for their children. However, unlike the female sex and younger age, a lower SES background was not found to be a risk factor to be classified in the high verbal abuse and emotional neglect and low household dysfunction class. It is likely that verbal abuse and emotional neglect are more commonly observed in Chinese rural families regardless of parental education level.

### 4.3. Latent Class Profiles Associated with Depressive Symptoms and Suicidal Ideation

Consistent with previous studies and the dose–response theory, our study further confirmed that participants in the high ACEs class and high verbal abuse and emotional neglect and low household dysfunction class were more likely to report depressive symptoms and more frequent suicidal ideation than participants in the low ACEs class. Meanwhile, participants in the high ACEs class also reported more depressive symptoms and suicidal ideation than those in the high verbal abuse and emotional neglect and low household dysfunction class. This finding is consistent with the dose–response theory, which suggests that students who experienced more ACEs were more likely to have adverse mental outcomes. Our study has also extended the literature on the association between ACEs and subsequent risk of increasing depressive symptoms (e.g., [5]) and suicidal behavior (e.g., [46]). In addition to the prolonged adverse impact of ACEs during childhood on future adults’ mental health demonstrated in previous studies (e.g., [47]), adversities during childhood also showed immediate impact on children’s mental health. Furthermore, although verbal abuse and emotional neglect were more commonly labeled as a normal parenting practice in rural China, compared with physical and sexual abuse being treated as evident trauma, our study highlighted that these types of obscure or less evident childhood trauma demonstrated a severe impact on children’s mental health. Particularly with the high prevalence of left-behind children in Chinese rural areas, emotional neglect became commonly observed due to the lack of parental figures to provide daily care.

## 5. Implications and Future Directions

Our study is one of the few studies that tried to explore ACEs among a large sample size of students in rural areas. In light of the findings, implications for the early identification and prevention of mental health problems and intervention strategies are proposed, particularly among female, younger, and low SES children with a history of ACEs. For example, when considering how to support the mental health of Chinese students in rural areas, practitioners could consider gathering information on students’ family settings and identifying any potential past family traumatic events. When working with a student, practitioners should try to understand how many ACEs a child has experienced and what types of ACEs they have experienced. At the same time, our findings also suggested the importance of raising the awareness of implicit abuse and neglect, such as verbal abuse and emotional neglect. On a community level, raising awareness of different types of ACEs and their detrimental outcomes among parents, teachers, and children in rural Chinese areas is also crucial to preventing the early onset of ACEs. For example, normalizing verbal abuse and emotional neglect should not be tolerated.

However, our study also has limitations that need to be addressed in future research. First, the participants were at a young age, with a mean of 10 years old and the youngest age 6 years old. It is likely that students at such an early age might not have been fully aware of the situations that happened at home when they were asked about household dysfunctions, such as whether there was anyone at home who was incarcerated or had mental disorders. At the same time, they might also have lacked awareness of abuse or neglect that happened to them, which results in underreporting. Future research suggests using a multi-informant approach or using questionnaires that ask more detailed items to understand younger students’ ACEs. Second, the study was cross-sectional. Thus, it is difficult to establish causation between variables. Nonetheless, our findings were still similar to those in previous longitudinal studies (e.g., [5]), which signifies the risk of ACEs on child mental health. Third, although we tested the model fitness with the goal of validating whether the measure was appropriate in the current sample, the four items that were dropped from the CES-DC were all reverse items. It was likely that the respondents might not have been highly attentive when filling out the questionnaire, which might lead to our results on depression needing to be interpreted with caution. Fourth, the wording of the suicidal attempt item may have led to response bias among young child participants due to direct wording. Further research could consider rephrasing the question in a more considerate manner. Fifth, to measure family SES, we did not use traditional questions, such as yearly household income, due to the young age of the participants. Instead, the participants were asked to rate whether they have certain appliances in their home that were commonly seen in middle-class households in rural China. Our measurement might not be the best representation of family SES. Future study is suggested to ask for caregivers’ input to gather SES information. Last but not least, there is a limitation in using the ACEs measure among children under 18 years of age. The definition of ACEs quantifies one’s exposure to traumatic events before the age of 18 [2]. Although previous studies have used the ACEs measure among children and adolescents (e.g., [34]), few have used the entire ACEs measure in their study. Future studies are suggested to conduct the statistical validation of the ACEs measure on their samples before ensuring its validity to use among child populations.

## 6. Conclusions

The present study filled a gap in the literature by attending to ACEs among a large sample of rural students in China. It was one of the first empirical studies that used a person-centered approach to advance the theoretical investigation of ACE profile patterns in the context of agism, sexism, and SES. In particular, the findings revealed that being female and younger and having a lower SES in rural areas were risk factors for students’ heightened ACEs, which implies the importance of advocating for gender, age, and SES equity in rural Chinese communities. Meanwhile, our identification of a unique latent class named high verbal abuse and emotional neglect and low household dysfunction suggested the importance of raising the awareness of implicit abuse and neglect in rural China, namely verbal abuse and emotional neglect. Lastly, the study contributed to the past literature and the dose–response theory on the harmful impacts of different magnitudes of ACEs on elementary school students’ mental health.

## Figures and Tables

**Figure 1 ijerph-19-16031-f001:**
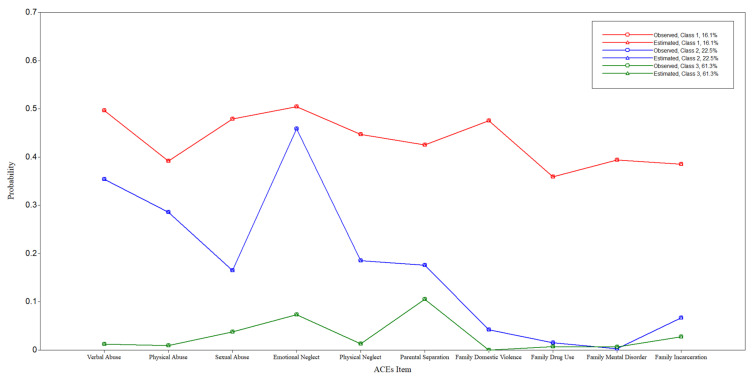
Three-Class Model.

**Table 1 ijerph-19-16031-t001:** Correlation Coefficients Among Variables.

	Age	Ethnicity	SES	Gender	Paternal Left-Behind	Maternal Left-Behind	Cumulative ACEs	Depression	Suicidal Ideation
Age	1								
Ethnicity	−0.60 ***	1							
SES	0.01	0.07 ***	1						
Gender	0.03 *	−0.02	0.02	1					
Paternal Left-Behind	0.08 ***	−0.04 **	−0.11 ***	0.03 *	1				
Maternal Left-Behind	0.07 ***	−0.055 ***	−0.07 ***	0.02	0.49 ***	1			
Cumulative ACEs	−0.13 ***	−0.03 *	−0.09 ***	0.12 ***	0.04 *	0.04 **	1		
Depression	−0.048 **	−0.027	−0.022	0.026	0.041 *	0.017	0.359 **	1	
Suicidal Ideation	−0.017	−0.003	0.001	0.048 **	−0.004	0.009	0.366 **	0.466 **	1

* *p* < 0.05,** *p* < 0.005, *** *p* < 0.001.

**Table 2 ijerph-19-16031-t002:** Fit indices of LCA Class Models.

Model (K-Class)	LL	npar	AIC	CAIC	BIC	saBIC	AWE	LRTS	BF (K, K + 1)
1-class	−16,704.608	10	33,429.22	33,503.36	33,493.36	33,461.59	33,607.51	–	0.000
2-class	−14,757.287	21	29,556.57	29,712.28	29,691.28	29,624.55	29,930.99	3894.64	0.000
3-class	−14,614.375	32	29,292.75	29,530.02	29,498.02	29,396.34	**29,863.29**	285.82	0.000
4-class	−14,518.423	43	29,122.85	**29,441.68**	**29,398.68**	29,262.04	29,889.51	191.90	36.487
5-class	−14,475.739	54	**29,059.48**	29,459.87	29,405.87	**29,234.28**	30,022.27	85.37	0.000

**Table 3 ijerph-19-16031-t003:** Log odds coefficients and odds ratio for the three-class model with demographic variables as covariates using the Class 3: Low ACEs class as the comparison group.

Disciplinary Techniques Class	Effect	Logit	*SE*	*T*	Odds Ratio
Class 1: High ACEs	Gender (reference group = male)	0.511 *	0.112	4.547	1.610
	Age	−0.326 *	0.057	−5.667	0.772
	SES	−0.145 *	0.031	−4.693	0.978
	Ethnicity	−0.023	0.033	−0.711	0.976
	Paternal Left-Behind Status	−0.143	0.128	−1.120	0.810
	Maternal Left-Behind Status	−0.224	0.136	−1.641	0.865
Class 2: High Verbal Abuse and Emotional Neglect and Low Household Dysfunction	Gender (reference group = male)	0.476 *	0.108	4.422	1.668
	Age	−0.259 *	0.056	−4.611	0.722
	SES	−0.022	0.031	−0.703	0.865
	Ethnicity	−0.024	0.032	−0.759	0.977
	Paternal Left-Behind Status	−0.210	0.131	−1.610	0.867
	Maternal Left-Behind Status	−0.145	0.140	−1.036	0.800

* *p* < 0.05.

**Table 4 ijerph-19-16031-t004:** Means and Mean Comparison of Depression and Suicidal Ideation in Each Class.

ACE Class	Depression (Range 1–4)	Suicidal Ideation (Range 1–4)
Class 1: High ACEs	2.40 (0.02) _a_	3.43 (0.02) _a_
Class 2: High Verbal Abuse and Emotional Neglect and Low Household Dysfunction	2.18 (0.03) _b_	1.29 (0.02) _b_
Class 3: Low ACEs	1.63 (0.03) _c_	1.06 (0.01) _c_

Note. Means that do not share subscripts differ at *p* < 0.001.

## Data Availability

The data presented in this study are available on request from the corresponding author.

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
