# Peer review of "The Latent Class Analysis of Adverse Childhood Experiences among Chinese Children and Early Adolescents in Rural Areas and Their Association with Depression and Suicidal Ideation"

_ijerph, 2022, doi:10.3390/ijerph192316031_

Round 1
Reviewer 1 Report
The aim of this review is to fill the gap in the ACEs literature to understand the ACEs patterns among Chinese children and early adolescents in rural areas and their subsequent experiences of depression and suicidal ideation. As specified, this was the first study that analyse this topic and can make a big contribution to deepening knowledge on the subject and applying, based on the results, measures to prevent negative episodes in children's lives. To highlight the use of a person-centered approach to examine ACEs in Chinese rural area students. The abstract presents the work appropriately. It also fully explains the methodology used and the objectives intended to be achieved. The introduction explains extensively and in detail the scientific background and rationale for the investigation. The use of dichotomous variables allows for a more streamlined analysis of the data but, more importantly, allows children to give direct answers without misinterpretation. The questionnaires used are consistent in assessing children's mental state. You have summarized the key findings with reference to the objectives of the study and compared with those in the literature, highlighting the strongest point, which is the first with this specific objective.
Author Response
Thank you for your review of our manuscript ijerph-2044181 “Latent Class Analysis of Adverse Childhood Experiences Among Chinese Children and Early Adolescents in Rural Areas and Its Association with Depression and Suicidality.” We strongly appreciate your acknowledgement and feedbacks.
Reviewer 2 Report
Thank you for inviting me to review this paper, which is an interesting and comprehensive study. I have only two suggestions for authors that might help strengthen their manuscript.
The authors provide very detailed information, but at times this can be excessive or unnecessary, so the work could be improved if it were more synthetic in some points, for example in the section “data collection”.
The description of the Depressive Symptoms and Suicidal Ideation measure should include literal examples of some items, as is done in the Adverse Childhood Experiences (ACEs) Questionnaire.
Author Response
Thank you for your review of our manuscript ijerph-2044181 “Latent Class Analysis of Adverse Childhood Experiences Among Chinese Children and Early Adolescents in Rural Areas and Its Association with Depression and Suicidality.” We have carefully considered the feedback and have summarized how we addressed each of the comments below. We also highlighted the revisions we made in the maintext using the “Track Changes” function. We appreciate all of the time and effort that you and the reviewers have put into this manuscript and feel it has been improved significantly through the revisions. Here is our point-to-point response.
- Thank you very much for pointing it out. We have shortened the data collection section on page 4. The reviewed section is as below.
“The present study was a part of a project that aimed to examine rural students’ psychological well-being and school functioning. Data collected was conducted in October 2018. First, the research team established collaboration with the rural elementary schools in Gansu Province, and 63 elementary schools agreed to participate in the project. Second, simple random sampling was conducted to select two random class cohorts from all 3rd- to 5th-grade classrooms. Parents were required to provide passive consent to agree for their child to participate in the study, and students needed to provide their assent before participation. The survey was distributed via paper-and-pencil format. Because the survey contained items that might provoke psychological discomfort, we provided the free local mental health hotlines to participants and parents on the informed consent. The study was conducted in accordance with the Declaration of Helsinki, and the protocol was approved by the Ethics Committee of IRB00001052-20020 in one of the correspondence authors’ university.”
- Thank you very much for your suggestion. Regarding the description of the literal examples of items. The revisions are addressed as below. For depressive symptoms measure, Sample items include “I was bothered by things that usually don’t bother me” and “I felt down and unhappy.” on line 238-239 on page 5 and, for suicidality item, The item was presented as “I have thought about attempting suicide.” on line 248-249 on page 5.
Reviewer 3 Report
Thank you for the opportunity to review this manuscript on latent class analysis of ACEs and association with depression and suicidality. The manuscript is excellent.
I recommend only the following minor changes:
- Title: replace suicidality with suicid ideation, as you only used 1 question on suicide ideation and therefore do not have data on suicidality, defined as all suicidal behaviour from thoughts to suicide
- Abstract, line 26: replace the word 'endorse' which is not appropriate and use instead 'have'
- Implications: asking 6 to 10 year olds 'how often have you thought about attempting suicide' is very direct and may lead to response bias. Consider adding for further research on asking about suicide in young ages to comment on possible response bias and to rephrase the question as: "Sometimes children think about hurting themselves or killing themselves when they are very upset. How many times have you thought about hurting yourself"
Author Response
Thank you for your review of our manuscript ijerph-2044181 “Latent Class Analysis of Adverse Childhood Experiences Among Chinese Children and Early Adolescents in Rural Areas and Its Association with Depression and Suicidality.” We have carefully considered the feedback and have summarized how we addressed each of the comments below. We also highlighted the revisions we made in the maintext using the “Track Changes” function. We appreciate all of the time and effort that you and the reviewers have put into this manuscript and feel it has been improved significantly through the revisions. Here is our point-to-point response.
- Thank you very much for your acknowledgement. We appreciate your time to review and helpful feedback.
- Thank you very much for pointing it out. We have revised the title and the term in the article. The changes are reflected in the title on line 4, keywords on lines 31-32, and maintext in line 281.
- Thank you very much for your suggestion. We agree that the original wording is too direct for young children. We have further added this as one of the limitations on lines 477-480 as below. “Fourth, the wording of the suicidal attempt item may lead to response bias among young child participants due to direct wording. Further research could consider rephrasing the question in a more considerate manner.”